# Extremely Rare Type of Breast Cancer—Dedifferentiated Breast Liposarcoma—Diagnosis and Treatment

**DOI:** 10.3390/jpm13101451

**Published:** 2023-09-29

**Authors:** Andrada-Elena Țigăran, Abdalah Abu-Baker, Daniela-Elena Ion, Teodora Peligrad, Daniela-Elena Gheoca-Mutu, Adelaida Avino, Andrei-Wilhelm Anghel, Andra-Elena Balcangiu-Stroescu, Anca Toma, Laura Răducu

**Affiliations:** 1Department of Plastic and Reconstructive Surgery, ‘Prof. Dr Agrippa Ionescu’ Clinical Emergency Hospital, 011356 Bucharest, Romania; tigaran.andra26@gmail.com (A.-E.Ț.); abdalah.abu-baker@drd.umfcd.ro (A.A.-B.); daniela-elena.ion@rez.umfcd.ro (D.-E.I.); teodora.peligrad@yahoo.com (T.P.); mutu.danielaa@gmail.com (D.-E.G.-M.); raducu.laura@yahoo.com (L.R.); 2Doctoral School, ‘Carol Davila’ University of Medicine and Pharmacy, 020021 Bucharest, Romania; 3Discipline of Anatomy, ‘Carol Davila’ University of Medicine and Pharmacy, 020021 Bucharest, Romania; 4Discipline of Plastic Surgery, ‘Prof. Dr Agrippa Ionescu’ Clinical Emergency Hospital, 011356 Bucharest, Romania; 5Department of Radiotherapy, Elias University Emergency Hospital, 011461 Bucharest, Romania; andrei.wilhelm@yahoo.com; 6Department of Radiotherapy, MedEuropa, 022343 Bucharest, Romania; 7Discipline of Physiology, Faculty of Dental Medicine, ‘Carol Davila’ University of Medicine and Pharmacy, 020021 Bucharest, Romania; stroescu_andra@yahoo.ro; 8Department of Anatomic Pathology, ‘Prof. Dr Agrippa Ionescu’ Clinical Emergency Hospital, 011356 Bucharest, Romania; toma.anca.irina@gmail.com

**Keywords:** sarcoma, breast cancer, dedifferentiated liposarcoma, radiotherapy

## Abstract

Primary liposarcoma of the breast is an uncommon soft tissue malignant tumor, comprising only 0.003% of all malignant breast tumors. The main differential diagnosis of this mass consists of malignant phyllodes tumor and metaplastic breast carcinoma. The objective of this paper is to report a case of dedifferentiated breast liposarcoma, therapeutic approach and outcome. We present a case of a 79-year-old woman complaining of a large mass in her left breast which had increased in size over the last 6 months. Physical examination revealed an enlarged left breast, and a total body CT scan showed a large tumor in contact with the musculature of the anterior thoracic wall, with no metastatic lesions. The histopathology report of a fine needle biopsy described a high-grade sarcoma. The Oncological Tumor Board recommended neoadjuvant radiotherapy sessions and reevaluation by MRI and CT scans. The patient underwent radical mastectomy with latissimus dorsi myo-cutaneous flap reconstruction. The final histopathology diagnosis was a grade 3 dedifferentiated liposarcoma (FNCLCC), with certain response to radiotherapy and positive MDM2, CDK4 markers. The postoperative period was uneventful; 12 months after surgery, the follow-up CT scan showed multiple pulmonary lesions with metastatic characteristics. Liposarcoma is a very rare type of breast cancer, and the most important treatment for breast sarcoma is surgery, the role of axillary lymph node removal, chemotherapy and radiotherapy still being controversial. Considering such cases are scarce and the development of surgical guidelines is difficult, reporting any new case is crucial.

## 1. Introduction

Primary breast sarcomas are uncommon, accounting for only 1% of all breast malignancies [1]. Liposarcomas are a common subtype of soft tissue sarcomas, the majority being diagnosed in the retroperitoneum and extremities, but the breast is an unusual site for liposarcoma development, primary breast liposarcoma (PBLS) representing 0.003% of all breast sarcomas [2]. PBLS almost exclusively affects women, generally affecting patients in their fourth to sixth decade, and clinical presentation is quite similar to that of other types of primary breast cancer [3]. The World Health Organization (WHO) has classified liposarcoma into seven subtypes—well-differentiated (WDLPS), dedifferentiated (DDLPS), myxoid, round cell, pleomorphic, mixed type and not otherwise specified [2].

Most patients with breast liposarcoma present with a single, firm, well-circumscribed, painless breast mass. Unlike benign breast tumors, patients often report a history of rapid growth over several months [4]. Using breast imaging can guide the diagnosis and, although ultrasonography and mammography can be useful, studies show that PBLS resembles fibroadenomas and abscesses, and can mimic phyllodes tumors in the early stages [5,6]. MRI and PET-CT are more dependable diagnostic tools for differentiating between these tumors. Also, PET-CT exhibits a sensitivity of 83.3% and a specificity of 85.7% in accurately identifying between WDLPS and DDLPS [7].

Considering that PBLS’ differential diagnosis includes breast carcinoma, malignant phyllodes tumor and, on clinical presentation, can mimic a fibroadenoma, the course of treatment depends on an accurate histopathology (HP) report. Diagnosis is confirmed using image-guided core biopsy. Fine needle biopsy is not recommended because it is inadequate in determining the subtype and grade of the lesion, which is crucial for the course of treatment and for the preoperative surgical planning [8]. Once the diagnosis of PBLS has been established, a whole body CT scan or MRI should be performed to rule out distant metastatic disease.

Complete surgical excision is of foremost importance and is considered the only potentially curative therapy for breast sarcomas. The purpose of surgery is to achieve the complete removal of the tumor with negative margins (R0). Axillary lymphnode removal does not improve overall survival (OS), and it is not recommended unless metastatic disease is suspected from imaging studies or if it is necessary in order to obtain tumor free margins [9]. Due to the rarity of this disease, there are no guidelines regarding adjuvant therapy, and the course of treatment has been adapted from the DDLS located in extremities or other sites. Adjuvant radiation is employed to reduce the risk of local recurrence in the case of high-grade DDLS of the extremity that is greater than 5 cm in diameter or after R1 resection that cannot be improved without causing major morbidity. Radiation planned in the neoadjuvant setting can help reduce the likelihood of local recurrence or in order to obtain a R0 resection. Because DDLS is relatively chemoresistant, systemic therapies are rarely employed for localized DDLS [10].

The prognosis of PBLS depends on several factors, including age, tumor size, grade and status of resection margins, but five years OS is still low, some studies showing between 20% and 55% survival rate [11,12].

## 2. Case Report

We present a case of a 79-year-old woman presenting in our department complaining of a large mass in her left breast. The patient revealed that she felt the mass upon self-breast examination a year ago, but over the last 6 months it has rapidly increased in size. The patient had no remarkable medical or family history, and she had no prior investigations or imagistic studies of the breast.

On physical examination the left breast was distended by a large, firm tumor of approximately 15 cm × 20 cm × 15 cm (Figure 1). The right breast revealed no pathological findings upon clinical examination, and no axillary lymph nodes or other changes were found.

We performed a full body CT scan which showed a large heterogenous cystic-like mass of the left breast, with intense iodophilic capsule and multiple septi and nodular components, of approximately 93/91/126 mm (Figure 2). The tumor was in close contact with the pectoralis muscle, both cranially and posteriorly, and the margins between the two structures were not well defined. The peritumoral breast tissue was dense and presented banding with important thickening of the left breast skin. Axillary and interpectoral lymph nodes were described, not exceeding 11/7 mm, with no malignant characteristics. Non-specific pulmonary micronodules were described in both lungs’ parenchyma.

An ultrasound-guided core biopsy was performed, and the HP report showed a sarcomatous cell proliferation containing approximately 80% necrosis in both specimens, with an abrupt transition from viable to non-viable tumor tissue. Elongated and polygonal cells with high pleomorphism and high mitotic rate: >20 mitoses/HPF were identified. The neoplastic cells displayed a purple nucleus and pink cytoplasmic clearing when colored with Hematoxylin and Eosin stain (Figure 3). The breast tissue was infiltrated by spindle tumor cells and inflammatory elements, with dilated vessels and glandular structures (Figure 4).

The immunohistochemical analysis was positive for vimentin, displaying tumor cells with brown cytoplasmic staining (Figure 5) and negative for bcl-2, actin, S100, CD34, keratin AE1/AE3 and estrogen receptors. Using the Fédération Nationale des Centres de Lutte Contre le Cancer (FNCLCC) grading system for soft tissue sarcoma, the patient was diagnosed with a grade 3 sarcoma.

After reviewing the case, the Oncological Tumor Board of “Agrippa Ionescu” Emergency Hospital, based on the patient’s data and using de FNCLCC guidelines, recommended radiotherapy prior to a total mastectomy, considering the tumors’ size, in order to achieve better resection margins.

The best treatment plan for this tumor was an Intensity-Modulated Radiotherapy- Volumetric Modulated Arc Radiotherapy (IMRT-VMAT) technique that targeted the specific volume of the left breast and regional lymph nodes (Figure 6). During a 4-week period, the patient received a TD (total dose) of 50Gy/25 fractions. The 2Gy/fraction, for 5 days a week, was supplemented by a 10Gy/5 fractions booster dose focused on the tumor (Figure 7).

After completing the radiotherapy regimen, the patient was admitted for reevaluation using thoracic MRI and full body CT scan. The MRI described a voluminous mass of about 12/14.3/14 cm present in the left breast, with predominantly fluid structure, a contrast-capturing capsule and pectoralis muscle invasion (Figure 8). No intrathoracic invasion was visible but there was possible focal tumor invasion on the surface of the IIIrd, IVth and Vth rib. No enlarged axillary lymph nodes were present on the MRI. The CT scan showed no secondary lesions.

A radical mastectomy of the left breast was performed, removing the tumor, breast tissue and pectoralis major and minor muscles (Figure 9). Upon visualization of the ribs, a segment of periosteum from the IIIrd and IVth rib was sent for extemporaneous examination. The results showed no tumor invasion, so the resulting defect (Figure 10) was covered by creating a latissimus dorsi myo-cutaneous flap. The postoperative period was uneventful, and the patient was discharged 2 weeks after the surgery.

The histopathology report described a 4.5 cm dedifferentiated liposarcoma ypT1 ypN0, with certain response to radiotherapy, comprising in 90% necrosis. Certain radiotherapy-induced changes were observed: extensive stromal hyalinization and the shrinkage of tumor cells, which confirmed the positive response to neoadjuvant therapy (Figure 11 and Figure 12).

All specimen margins were negative (closest margin-deep fascia 0.05 cm). The immunohistochemical analysis was negative for keratin (AE1/AE3), cytokeratin 5/6, p63, p16, S100, CD34, actin, desmin, ERG, SOX10 and positive for CDK4 (Figure 13) and MDM2 (mouse double minute 2—Figure 14), in both immunohistochemical analysis neoplastic cells displaying brown nuclei compared with the control cells.

Because DDLS can mimic any type of sarcoma, MDM2 immunohistochemistry/FISH should be part of the panel of tests in any sarcoma occurring at any site, certainly if it is undifferentiated and pleomorphic, and so a FISH test to identify MDM2 was recommended. The test came back positive for overexpression of the MDM2 gene.

No further treatment was recommended for the patient. Evaluation every 2 months and CT scans every 3 months in the first year were recommended. Twelve months after surgery, the patient showed completely healed donor and recipient site (Figure 13), with a good quality of life, but the control CT scan revealed multiple lesions on both lungs, with metastatic characteristics. The patient is presently (Figure 15) evaluated for chemotherapy with doxorubicin.

## 3. Discussion

PBLS is a very rare disease, and it is difficult to identify specific treatment for these patients. The vast majority of publications regarding this topic are case reports and single institution retrospective analyses, which makes the development of clear protocols a real challenge.

There is a widespread acceptance that surgical resection should be the first-line treatment for BS. The largest study to date, which included 20 patients with PBLS and up to 14 years of follow-up, was conducted in 1986 [13]. They concluded that the best course of treatment for PBLS is surgery with negative excision margins R0. For many years, mastectomy was considered the gold standard and the best method to obtain excellent local control of the disease. Some studies show significantly increased OS after total mastectomy compared to conservative surgery in PBS patients [12]. In contrast, Yin et al. recommended breast-conserving surgery (BCS) if complete resection with negative margins can be achieved, and adjuvant radiotherapy if the tumor exceeds 5 cm in size, thus concluding that mastectomy plus radiation versus mastectomy showed better OS, favoring combination treatment in T2M0 tumors [14]. In choosing between mastectomy and BCS, it is important to evaluate the tumor size and tumor/breast size ratio before surgery in order to properly perform a correct planning of the surgical strategy which could include even BCS without compromising the prognosis [15].

Considering the rarity of PBLS, the present course of treatment mirrors smooth tissue sarcomas’ (STS) protocols. Current recommendations for PBS treatment are derived from small retrospective studies and case reports and extrapolated from non-breast STS. By evaluating the results from STS and PBS studies, we can better understand the disease and the treatment choices. A multicenter, randomized, phase 3 trial analyzed the difference between surgery versus preoperative radiotherapy and surgery for patients with primary retroperitoneal sarcoma, a type of STS, and concluded that in the liposarcoma cohort specifically, twice as many local relapses were observed in the surgery group than in the radiotherapy plus surgery group, possibly related to the impact of radiotherapy [16].

DDLPS tends to have poor chemosensitivity, but recent studies suggest that selected high-risk patients with STS and an OS of 51% or less may benefit from adjuvant chemotherapy [17]. Pasquali et al. aimed to determine whether the patients with high-risk STS benefited from adjuvant chemotherapy in the EORTC-STBSG 62931 [18] randomized controlled trial which failed to detect an impact for doxorubicin plus ifosfamide, and using the prognostic nomogram Sarculator, they identified three groups based on the pr-OS (predicted-OS). The patients in the low pr-OS had better outcomes when treated with adjuvant chemotherapy. Currently, a phase III multicenter international randomized trial, with results anticipated in 2028, is determining whether neoadjuvant chemotherapy followed by surgery or surgery alone impacts the disease-free survival of patients with high-grade retroperitoneum DDLS or leiomyosarcoma [19].

In the context of metastatic DDLS, the recommended first-line therapy is an anthracycline-based chemotherapy regimen. Doxorubicin is currently the treatment of choice. In the phase III trial EORTS 62012, patients with advanced or metastatic STS were randomized to doxorubicin monotherapy or doxorubicin-ifosfamide combination therapy. No statistically significant improved OS was observed; even though higher response rates and progression-free survival were noted, there were more toxicity-related events in the combination therapy group [20].

Although not specific to WDLPS/DDLPS, MDM2 amplification is seen in nearly 100% of DDLPS and WDLPS [21]. MDM2 and/or CDK4 protein overexpression and gene amplification are beneficial ancillary studies that can help establish the diagnosis of primary breast ALT/WDLPS and DDLPS, and effectively rule out the diagnoses of malignant phyllodes tumor and metaplastic breast carcinoma [22]. Both WDLPS and DDLPS contain high-level amplifications of chromosome 12q13-15; this region includes several genes, including CDK4 and MDM2. MDM2 is consistently amplified and overexpressed, and it is considered to represent one of the earliest events in the formation of WDPL/DDLPS [23]. MDM2 is considered to be the main negative regulator of p53, the tumor suppressor gene most frequently mutated in human cancers, preventing nuclear translocation and transcription and promoting its degradation by E3 ubiquitin ligase. MDM2 amplification impairs the apoptotic activity of p53, causing tumor proliferation, and by overexpressing MDM2, cancer cells have another means to block p53. The overexpression of MDM2 in DDLPS is linked with worse prognosis and poor response to chemotherapy and so MDM2-p53 interaction has garnered interest as a therapeutic target for DDLPS and other malignancies [24]. Bill et al. showed that MDM2 inhibition with SAR405838, a non-Nutlin small molecular inhibitor of MDM2, induced cell-cycle arrest and apoptosis in wild-type p53 human DDLPS tumor cells, which was associated with the enrichment of p53-mediated gene expression patterns. The authors also showed that SAR405838 was associated with significant reductions in tumor volume in DDLPS xenograft-bearing mice [25]. In a phase I trial for SAE405838, in which patients received SAR405838 monotherapy, a second cohort composed of 21 patients with DDLPS was tested to determine the maximum tolerated dose. The patients within the DDLPS cohort had a 56% rate of stable disease- and progression-free survival at 3 months was 32%. The authors concluded that despite the lack of objective response, evidence of disease stabilization and p53 activation in the majority of patients warranted further evaluation, particularly for combination regimens [26].

DS3032b, a MDM2 inhibitor, was evaluated in a phase I study of patients with WDLPS/DDLPS, solid tumors, and lymphomas. Ninety-four patients were enrolled in the study, 60% of patients achieved stable disease for a median duration of 6.7 months, with three partial responses observed among patients with DDLPS, synovial sarcoma, and squamous cell lung cancer. Adverse events included thrombocytopenia (61%) and neutropenia (28%), with 8% of patients experiencing dose-limiting toxicities [27].

Dickson et al. conducted a phase II, non-randomized clinical trial in which patients with advanced WDLPS and DDLPS received palbociclib, a CDK4/6 inhibitor. Of the 28 patients, 57% achieved progression-free survival at 12 weeks, and one patient achieved durable complete response 2 years after treatment. The authors did not include histology-specific outcomes and close to 80% of patients experienced tumor growth while on treatment [28].

## 4. Conclusions

Breast sarcoma is a rare form of breast cancer that generally exhibits similar biological behavior to sarcomas in other parts of the body, rather than epithelial breast malignancies, like carcinoma. 

Lymph node metastases are uncommon in breast sarcomas, so routine lymphadenectomy is generally not recommended. However, adjuvant radiation and/or chemotherapy may be considered for high-risk patients to improve local control and treat any potential subclinical metastatic disease, especially in the case of DDLPS, which has a high risk of local recurrence. Currently, the focus is on defining sarcomas based on their histologic subtype, molecular profile and genetic changes, rather than their anatomic origin, which will aid the development of novel therapies, like MDM2 and CDK4/6 inhibitors.

## Figures and Tables

**Figure 1 jpm-13-01451-f001:**
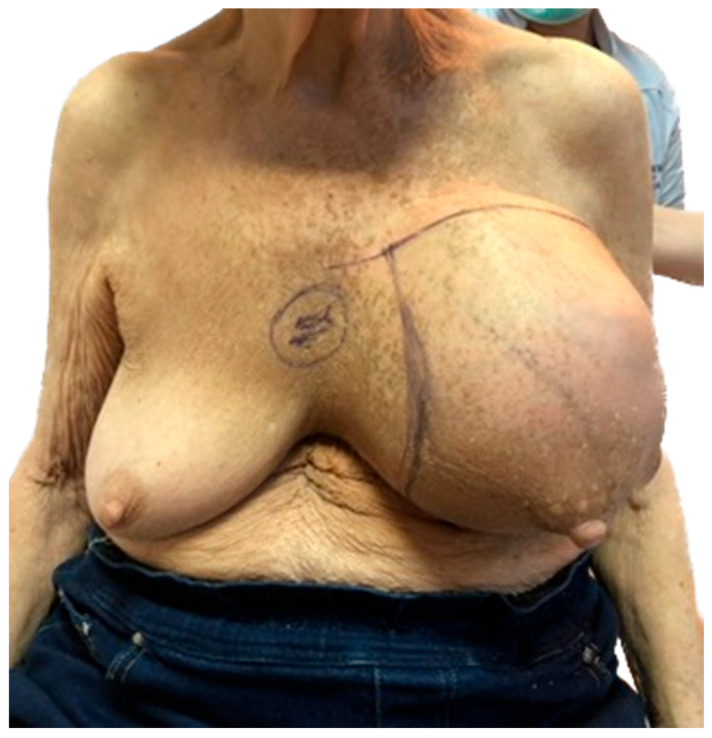
Patient presenting with a distended left breast; first examination.

**Figure 2 jpm-13-01451-f002:**
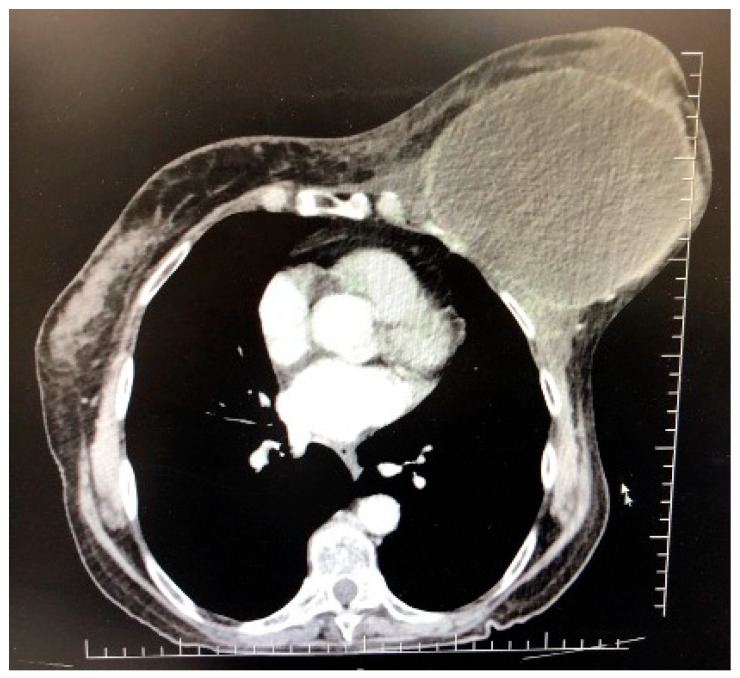
CT scan showing a well-circumscribed large mass of the left breast, in close contact with the pectoralis major muscle.

**Figure 3 jpm-13-01451-f003:**
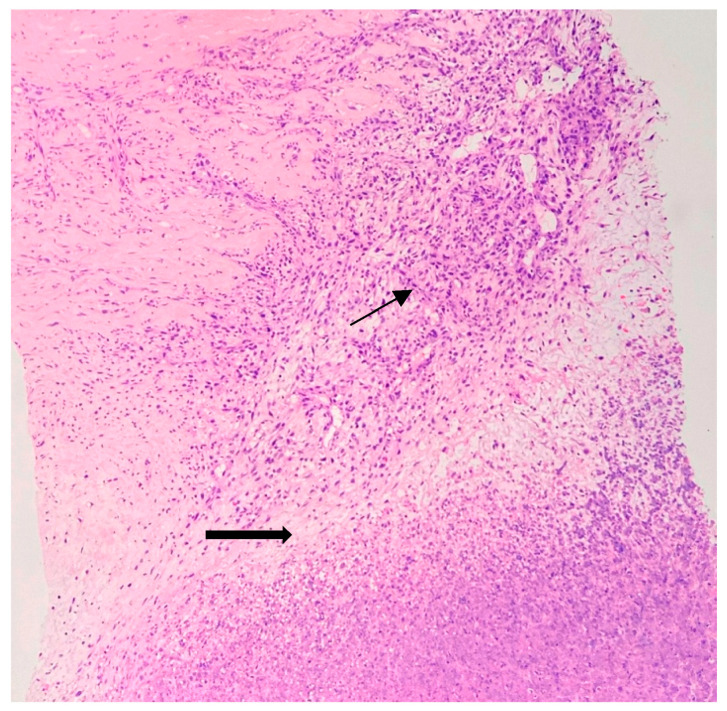
Initial biopsy. Some neoplastic cells (purple nucleus and pink cytoplasm with focal cytoplasmic clearing—thin arrow) in myxoid-like areas with distinct stromal vessels. Abrupt transition from viable to nonviable tumor with large area of coagulative necrosis and karyorrhectic debris—thick arrow (Hematoxylin and Eosin stain, 10× objective).

**Figure 4 jpm-13-01451-f004:**
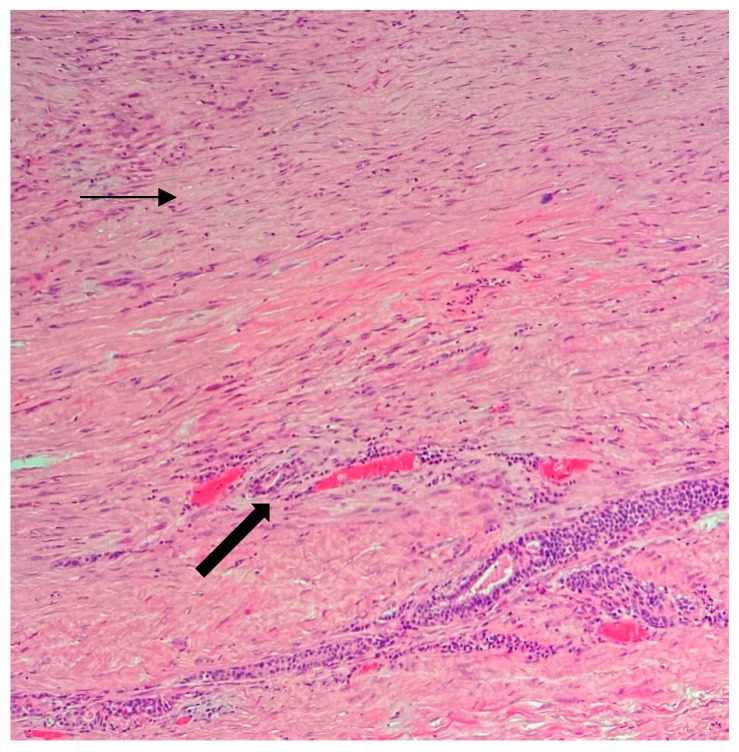
Initial biopsy. Upper part: spindle tumor cells (purple nucleus and pink abundant cytoplasm—thin arrow) that are loosely fascicular with regions of sclerotic matrix and a chronic inflammatory component. Lower part: breast parenchyma with glandular structures and dilated vessels—thick arrow (Hematoxylin and Eosin stain, 10× objective).

**Figure 5 jpm-13-01451-f005:**
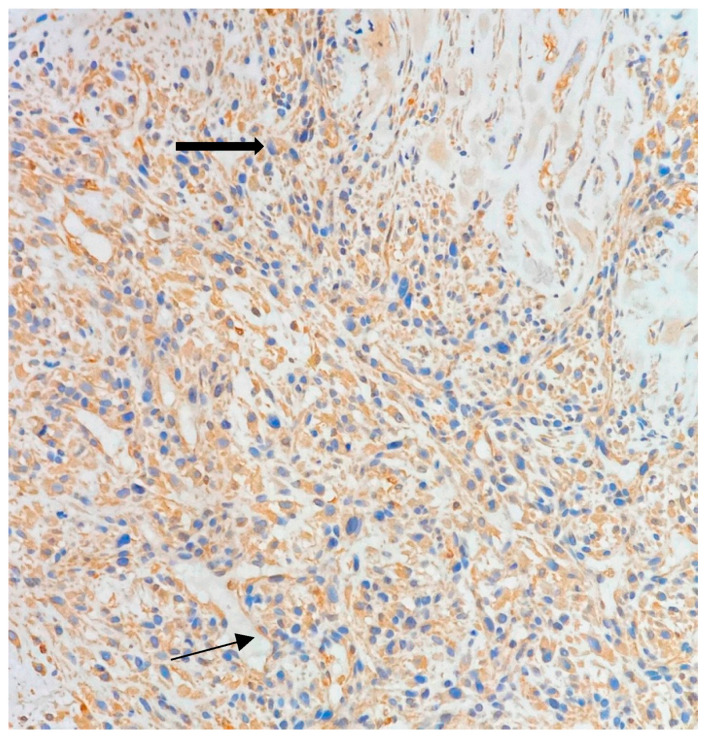
Initial biopsy. Cytoplasmic staining (brown) of tumor cells confirming mesenchymal origin—thick arrow. Endothelial cells (lower left of the image) also stain positive and represent the positive internal control—thin arrow (Vimentin stain, 20× objective).

**Figure 6 jpm-13-01451-f006:**
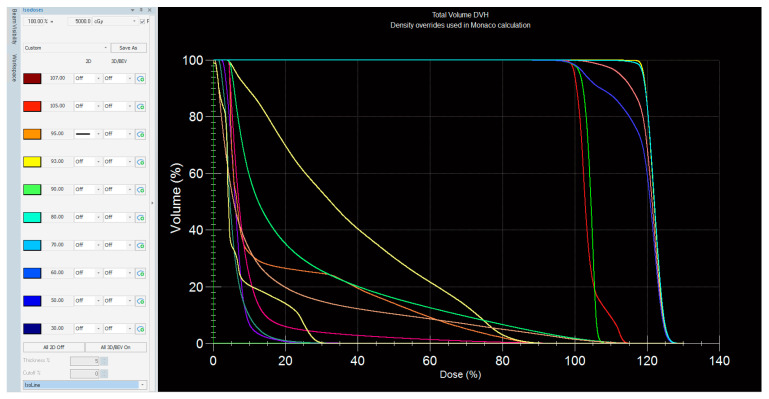
The dose per volume calculations used in IMRT-VMAT radiotherapy regimen.

**Figure 7 jpm-13-01451-f007:**
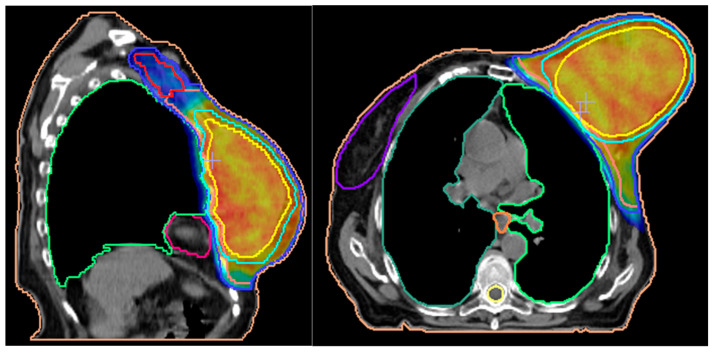
3D planning for external radiotherapy protocol used in our patients’ case; the highest dose was focused on the tumor volume. The scale used to measure the radiation dose required for each site was color-charted from blue (low dose, 4750 cGy) to red (highest dose- 6540 cGy).

**Figure 8 jpm-13-01451-f008:**
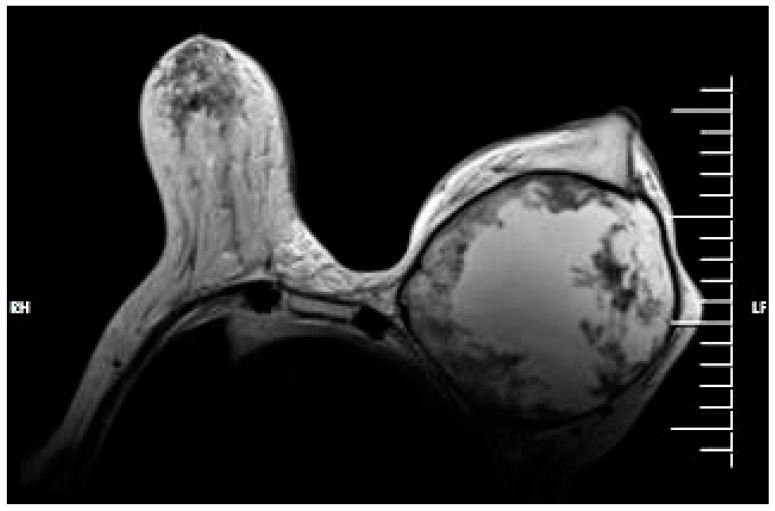
Contrast-enhanced MRI showing a cystic-like mass distending the left breast.

**Figure 9 jpm-13-01451-f009:**
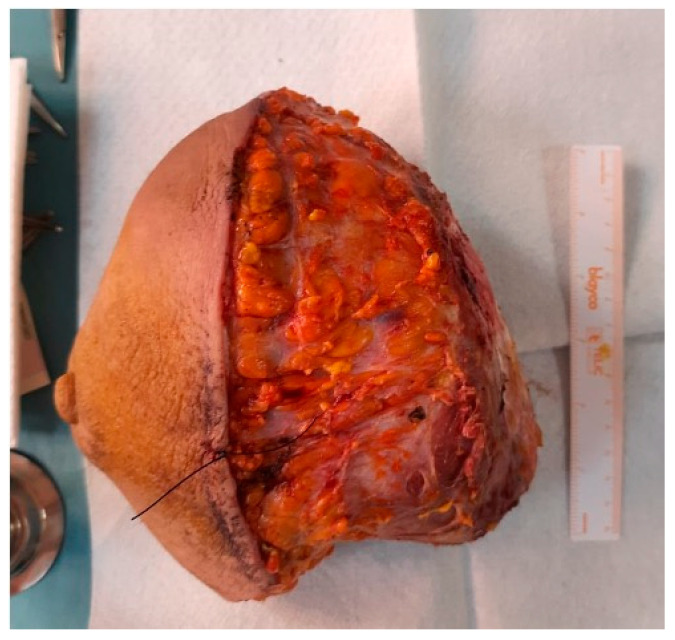
Radical mastectomy specimen (1950 g).

**Figure 10 jpm-13-01451-f010:**
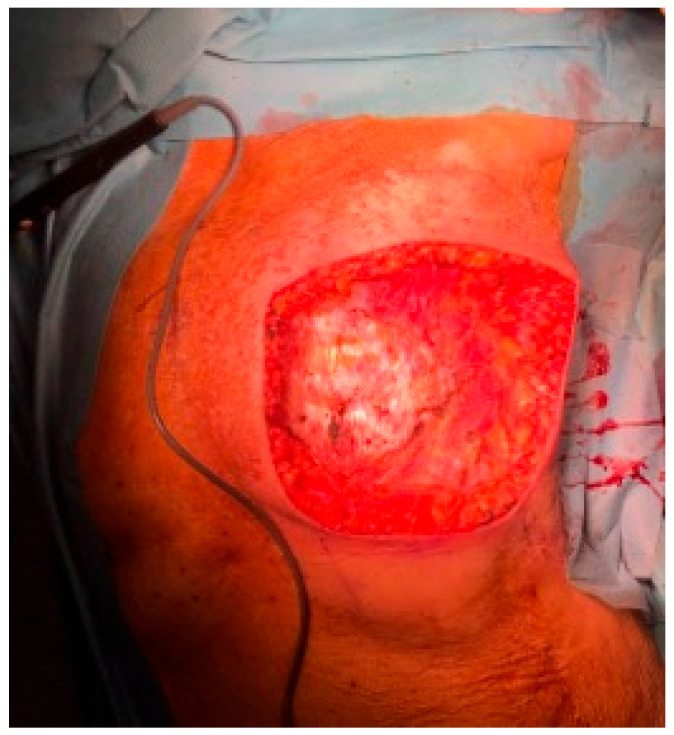
Post-mastectomy defect.

**Figure 11 jpm-13-01451-f011:**
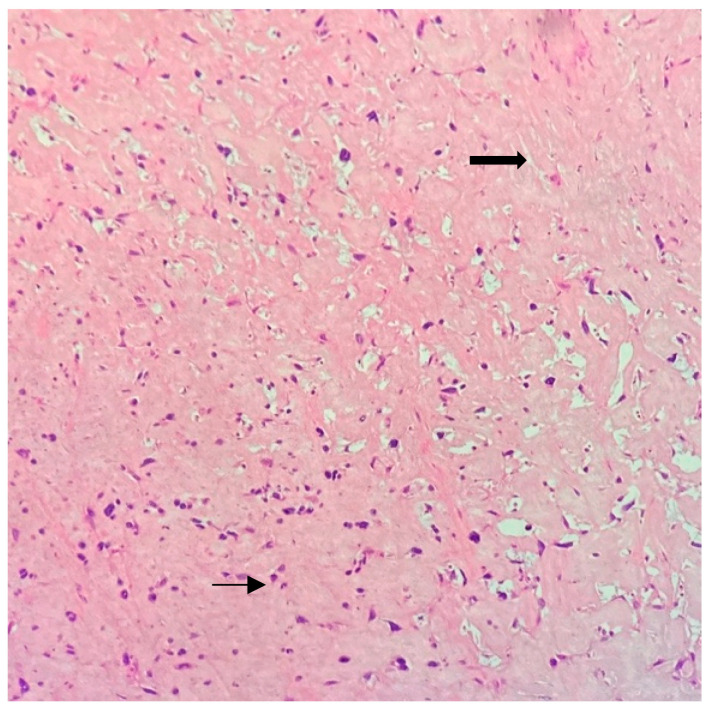
Neoadjuvant radiotherapy changes with extensive stromal hyalinization (bright pink—thick arrow) and shrinkage of tumor cells (purple—thin arrow). (Hematoxylin and Eosin stain, 20× objective).

**Figure 12 jpm-13-01451-f012:**
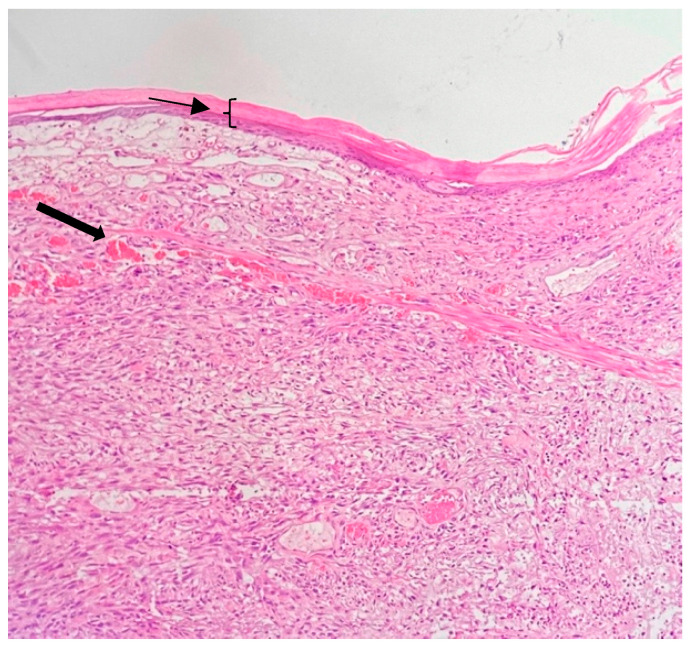
Radiotherapy-associated alterations with epidermal atrophy (thin arrow), dermal oedema and telangiectatic vessels (thick arrow). Viable tumor cells (purple nucleus and pink cytoplasm) and necrotic debris (purple) are present in the lower dermis. (Hematoxylin and Eosin stain, 20× objective).

**Figure 13 jpm-13-01451-f013:**
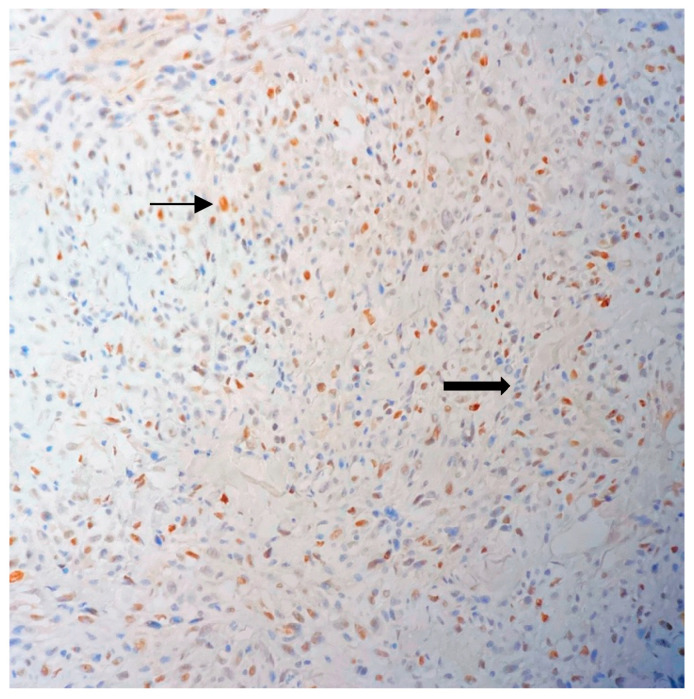
Immunohistochemical analysis: Most neoplastic spindle cell nuclei (brown) are positive for CDK4—thin arrow. Non-neoplastic fibroblasts and endothelial cells represent the negative internal control with absence of staining—thick arrow. (CDK4 Immunohistochemistry, 20× objective).

**Figure 14 jpm-13-01451-f014:**
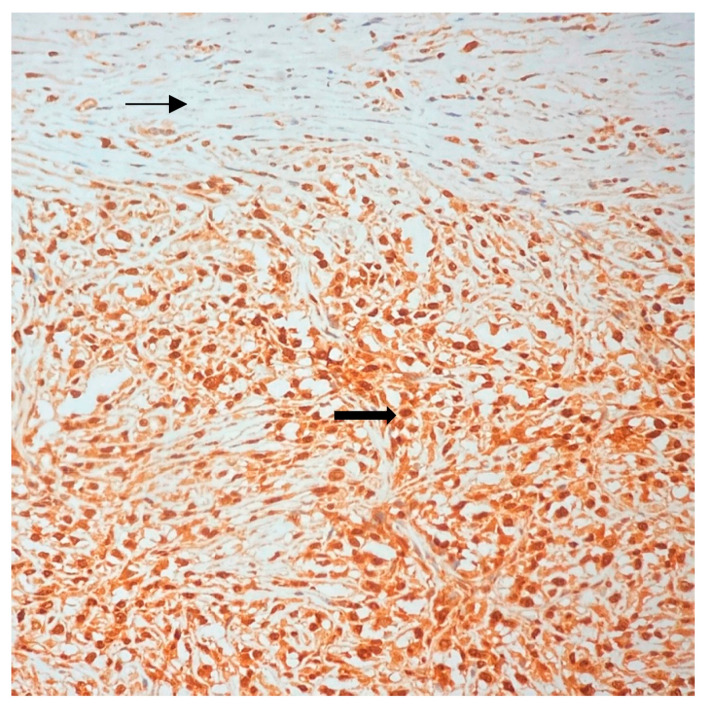
Immunohistochemical analysis: All tumor cells display nuclear (brown) positivity for MDM2—thick arrow. Non-neoplastic fibroblasts (upper part of the image) represent the negative internal control and do not stain—thin arrow. (MDM2 Immunohistochemistry, 20× objective).

**Figure 15 jpm-13-01451-f015:**
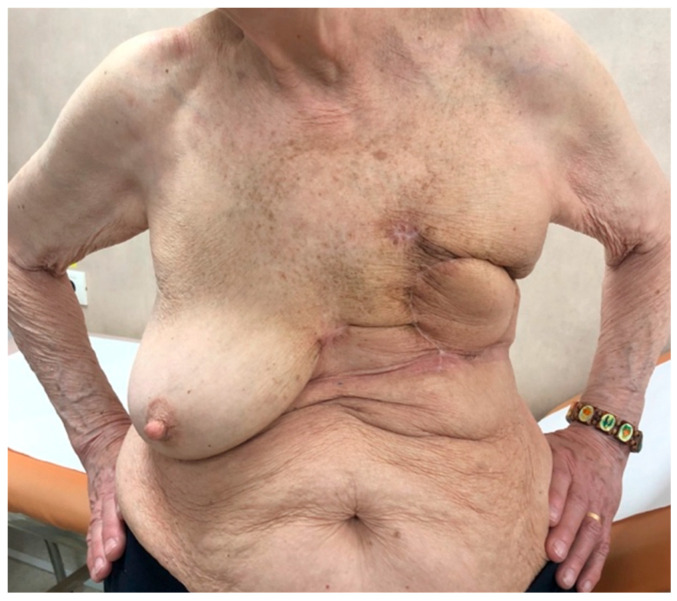
Twelve months after mastectomy surgery, fully healed recipient site.

## Data Availability

No new data were created.

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
