# Peer review of "Extremely Rare Type of Breast Cancer—Dedifferentiated Breast Liposarcoma—Diagnosis and Treatment"

_jpm, 2023, doi:10.3390/jpm13101451_

Round 1

Reviewer 1 Report

Extremely rare type of breast cancer - dedifferentiated breast liposarcoma-diagnosis and treatment

The main objective of this work was to report a case of dedifferentiated liposarcoma of the breast, therapeutic approach and evolution.

Minor comments

* I suggest that in the biopsy stains specify what is observed in each color.

* Are there comparative controls from sites adjacent to the tumor? Is there expression of any of the markers evaluated in these sites adjacent to the tumor?

* Have blood markers been described that help detect Liposarcoma?

Author Response

Thank you for the annotations and the review provided.

We have added a detailed description of the biopsies to provide a better understanding of the histopathological aspect described. Additionally, adjacent control markers to the tumor have been identified and described. As far as we are aware, there are no blood markers to diagnose liposarcoma.

Should we modify something else?

Best regards, 
Adelaida Avino

Reviewer 2 Report

 In this article, the authors described a case report of a 79-year-old woman with dedifferentiated breast liposarcoma- The patients had a large mass in her left breast and physical examination revealed an enlarged left breast and a total body CT-scan showed a large tumor in contact with the musculature of the anterior thoracic wall, with no metastatic lesions. The histopathology report of a fine needle biopsy described a high-grade sarcoma. The patient underwent radical mastectomy with latissimus dorsi myocutaneous flap reconstruction. Final histopathology diagnosis was a grade 3 dedifferentiated liposarcoma (FNCLCC), with certain response to radiotherapy and positive MDM2, CDK4 markers. The postoperative period was uneventful; 12 months after surgery, the follow-up CT scan showed mutiple pulmonary lesions with metastatic characteristics. They concluded that  the cases are carce and the development of surgical guidelines is difficult, reporting any new case is crucial. The paper is interesting

Author Response

Thank you very much for your nice comment. 

Best regards, 

The team from Bucharest